
# Asymmetric transport in long-range interacting chiral spin chains

**Javad Vahedi**[1,2][⋆]

**1** Technische Universität Braunschweig, Institut für Mathematische Physik,
Mendelssohnstraße 3, 38106 Braunschweig, Germany
**2** Department of Physics and Earth Sciences, Jacobs University Bremen,
Bremen 28759, Germany

⋆ j.vahediaghmashhadi@tu-bs.de

## Abstract

Harnessing power-law interactions ($1/r^\alpha$) in a large variety of physical systems are increasing. We study the dynamics of chiral spin chains as a possible multi-directional quantum channel. This arises from the nonlinear character of the dispersion with complex quantum interference effects. Using complementary numerical and analytical techniques, we propose a model to guide quantum states to a desired direction. We illustrate our approach using the long-range XXZ model modulated by Dzyaloshinskii-Moriya (DM) interaction. By exploring non-equilibrium dynamics after a local quantum quench, we identify the interplay of interaction range $\alpha$ and Dzyaloshinskii-Moriya coupling giving rise to an appreciable asymmetric spin excitations transport. This could be interesting for quantum information protocols to transfer quantum states, and it may be testable with current trapped-ion experiments. We further explore the growth of block entanglement entropy in these systems, and an order of magnitude reduction is distinguished.

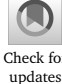

# 1 Introduction

Recently, long-range interacting quantum systems have received an increasing attention in quantum applications [1–5]. Many natural and engineered quantum systems show long-range interactions decaying with distance $r$ as a power law $1/r^\alpha$, such as van der Waals or dipole-dipole interactions. This has triggered fundamental questions related to the spreading of correlation in such systems, in particular the generalization of light-cone picture of the Lieb-Robinson (LR) bound [6]. Comprehension the quantum dynamics in this more general case is an active field of theoretical research [7–17].

With local quenches in the long-range transverse Ising model [3, 7] and XY model [4], the behavior of correlation propagation is classified into different regimes as a function of the decay exponent $\alpha$. For spreading of correlation, the dynamics are divided into (i) a regime of short-range interactions where $\alpha > 2$, and (ii) a regime of intermediate- and long-range interactions when $\alpha < 2$, with certain features also changing at $\alpha = 1$. While the light-cone picture remains a good description for short and intermediate-range, at long-range the maximal velocity is predicted to diverge and the light-cone is no longer exists.

Moreover, noncollinear spin models with unique rotational states, such as chiral spin-spirals, have also received great attention for their application potential in spintronics, information technology and likelihood hosts for Majorana Fermions with coupled to a superconductor [18, 19]. Moreover, experiments show the importance of Dzyaloshinskii-Moriya (DM) interactions in the appearance of a spin-spiral ground state configuration [20] and the appearance and importance of a vector spin chirality, an order parameter, in the system dynamics and transmit information down the chain [21, 22]. While linear coupling in the spin models ($\sim S_i \cdot S_j$) favors a parallel (ferromagnetic) or antiparallel (antiferromagnetic) alignment, the DM coupling ($\sim S_i \times S_j$) favors perpendicular alignment, which could possess a frustrated ground state [23], as well as rich three-dimensional spin skyrmion structures [24]. Furthermore, long-range DM interactions showed a rich phase diagram and quantum dynamics in 1D systems [25]. The DM couplings between spins exist only when the inversion symmetry is broken at the middle point between the two spins [26, 27]. For models with inversion symmetry, the external electric field $\vec{E}$ induces the DM interaction. Namely $D_{ij} \sim \vec{E} \times \vec{e}_{ij}$, where $\vec{e}_{ij}$ is the unit vector connecting the two sites $i$ and $j$.

Motivated by these possibilities, we study local quench dynamics in the long-range interacting XXZ model [3, 4] modulated with noncollinear DM interaction. While several studies on 1D systems with colinear interactions have been reported, the effect of long-range DM interactions has not been explored before (to the best of our knowledge). Using complementary numerical and analytical techniques we explore the possible quantum state engineering, and our main results on the quantum correlation spreading are summarized in Fig.1. It sketches the situation when a polarised spin state undergoes a local quench dynamics when long-range DM interaction is absent or present. In the absence of DM interaction, excitation propagates symmetrically about the perturbation. While the presence of DM interaction modulates an asymmetric transport for all $\alpha$'s, but is only appreciable for intermediate- and long-rage (small $\alpha$). The direction in which spin excitation transport can also be controlled by DM coupling. A similar asymmetric transport has been recently reported in Abelian anyons [28]. Contrary to the Abelian anyons in which non-local anyonic commutation plays a role, in fermionic and bosonic models presented in this work complex quantum interference induces asymmetric transport.

The rest of this article is arranged as follows: In Sec. II, we present the long-range models, the quantum quench protocol setup, and details of the tools we use to probe the experiment. In Sec. III, we then present the results, giving details of the methods we use to solve. We provide a summary in Sec .IV.

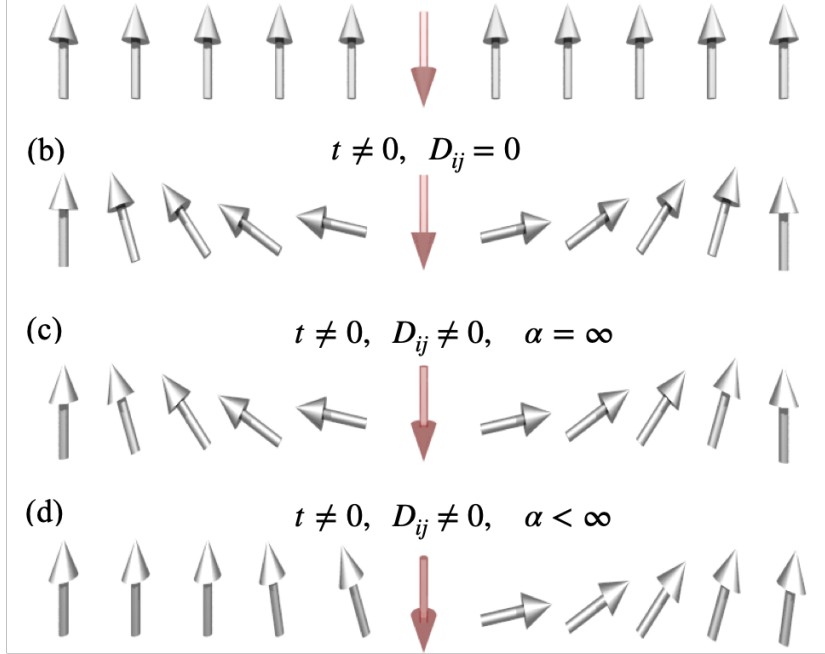

Figure 1: (colour online) Sketch of the protocol. (a) at time $t = 0$ initializing state with all spins aligned in the $z$-direction, and perturb single spin at the middle of chain by reversing it direction. Probing local spin evolution at later times (b) in the absence of DM interaction $D_{ij} = 0$ for all range of interaction, and in the presence of DM interaction $D_{ij} \neq 0$ for (c) nearest-neibour interaction $\alpha = \infty$, (d) at finite $\alpha < \infty$ but significant in long-range regime $\alpha \to 0$.

## 2  Models and Observables

Two long-range interacting Ising [3] and XY [4] spin models have already been engineered experimentally with ion trapped technique and studied theoretically [7]. The out-of-equilibrium dynamic following local and global quenches in these two models have been investigated, and shown that at sufficiently long-range interaction, the locality is expected to be spoiled (namely, not obeying LR bound). Here, we assume it could also be feasible to engineer a long-range interaction as of DM interaction which occurs when a model is no longer symmetric under inversion [26, 27]. Having this assumption, we come up with the following long-range XXZ Hamiltonian:

$$\mathcal{H} = \sum_{i \neq j} \Big[ J_{ij} \left( \sigma_i^x \sigma_j^x + \sigma_i^y \sigma_j^y + \Delta \sigma_i^z \sigma_j^z \right) + D_{ij} \cdot \left( \boldsymbol{\sigma}_i \times \boldsymbol{\sigma}_j \right) \Big] + h \sum_i \sigma_i^z, \tag{1}$$

where $J_{ij} = J/|i - j|^\alpha$ and $D_{ij} = D\hat{z}/|i - j|^\alpha$ with tunable exponent between infinite range ($\alpha = 0$) and nearest-neighbour ($\alpha = \infty$), $J$ and $D$ denote the nearest-neighbor interaction strength, $\Delta$ is exchange anisortiopy, and $h$ is local magnetic field. In this system, the total axial magnetization $S_{\text{total}} = \sum_{i=1}^L \sigma_i^z$ is a conserved quantity. Model is nonintegrable for all finite $\alpha$, but reduces to integrable models in the limit $\alpha \to 0$.

Local and global quenches are two well-tested protocols in the community of nonequilibrium dynamics. Here, we follow exactly the local quench setup as used in the experiment of Ref. [3]. We prepare a polarised spin state in which all spins are aligned in the external field $z$-direction (see Fig.1-a), as a ground state of the models at extreme case $h \gg \max\{J_{ij}, D_{ij}\}$, then

we perturb a single spin at the middle of the system by flipping its direction $|\psi_0\rangle = \sigma_{L/2}^x |\psi_{GS}\rangle$, as $|\psi_0\rangle = |\uparrow \cdots \uparrow\downarrow\uparrow \cdots \uparrow\rangle$, and observing its subsequent evolution. This can be done by probing spatially and temporally resolved spin polarization $\langle\sigma_i^z\rangle_t$ or equal-time connected correlation function $C_{i,L/2}(t) = \langle\sigma_i^z\sigma_{L/2}^z\rangle_t - \langle\sigma_i^z\rangle_t\langle\sigma_{L/2}^z\rangle_t$.

The block entanglement entropy (EE) also gives interesting information of quantum correlations between two segments of a system (namely the left and right half of the chain). We will use the von Neumann entropy, which is given via

$$S_{vN}(\rho_l) \equiv S_l \equiv -\text{tr}(\rho_l \log \rho_l), \tag{2}$$

where $\rho_l$ is reduced matrix of the left segment of chain and is defined as $\rho_l \equiv \text{tr}_r(|\psi\rangle\langle\psi|)$, which $\text{tr}_r$ denotes the partial trace over the right segment of chain. In this work, we consider the EE of half of the chain $S_{L/2}(t)$, as its time-dependent growth summaries the buildup of quantum correlations between two halves of the chain.

Spreading information and distribution of entanglement across either side of the initial excitation in this setup could potentially be useful for application. Quantum mutual information, which gives more information on the distance of correlations [8], is the information between two distant spins $i$ and $j$, and is defined via

$$\mathcal{I}_{ij} = S_{vN}(\rho_i) + S_{vN}(\rho_j) - S_{vN}(\rho_{ij}), \tag{3}$$

where $\rho_i = tr_{k\neq i}(|\psi\rangle\langle\psi|)$ and $\rho_j = tr_{k\neq j}(|\psi\rangle\langle\psi|)$ denote the reduced density matrices of the single spins (obtained by tracing over all other spins $k$), and $\rho_{ij} = tr_{k\neq i,j}(|\psi\rangle\langle\psi|)$ is the reduced density matrix of the composite system of the two spins.

As said before, our main goal is to explore the way correlations propagate in the model and plausible control over them. In the following section, we exploit both analytical and numerical techniques to measure the above tools to tackle the problem.

## 3 Results

To get an intuition, it may help to start with the nearest-neighbor (namely $\alpha = \infty$) model with $\Delta = 0.0$ and map Hamiltonian Eq.(1) to a fermionic model. This model can be solved analytically using the Jordan-Wigner transformation. For the long-range interacting case, the analytical fermionic picture is no longer applicable, so one can switch to the bosonic language as an analytical tool or exploit numerical techniques to inspect the dynamic of the model. To treat the many-body problem and considering the interacting term $\Delta \neq 0$, we choose the exact diagonalizing (ED) method but note that ED is limited to small system size $L \approx 14$ due to exponential growth of the Hilbert space dimension $2^L$. However, to study the dynamic of many-body systems, the Krylov-space technique with exploiting the sparse structure of Hamiltonian can push limit bigger system size [29–31]. We notice for the long-range interaction the Hamiltonian matrix is not sparse as a short-range case. Nevertheless, with the method we were able to reach system sizes up to $L = 23$ with dimension $\dim(\mathcal{H}) = 2^{23} \approx 10^7$.

### 3.1 Mapping to fermionic particles ($\Delta = 0.0$)

To understand the excitation spread, it is instructive to start with the case of nearest-interactions, i.e., with a decay exponent $\alpha \to \infty$, and discuss the dynamics of the quantities in this regime. In the limit $\Delta = 0.0$, the model Hamiltonian Eq.(1) becomes a standard XX model of the form

$$\mathcal{H}_{fermion} = \sum_j \tilde{J}\left(\sigma_j^x\sigma_{j+1}^x + \sigma_j^y\sigma_{j+1}^y\right) + h\sigma_j^z, \tag{4}$$

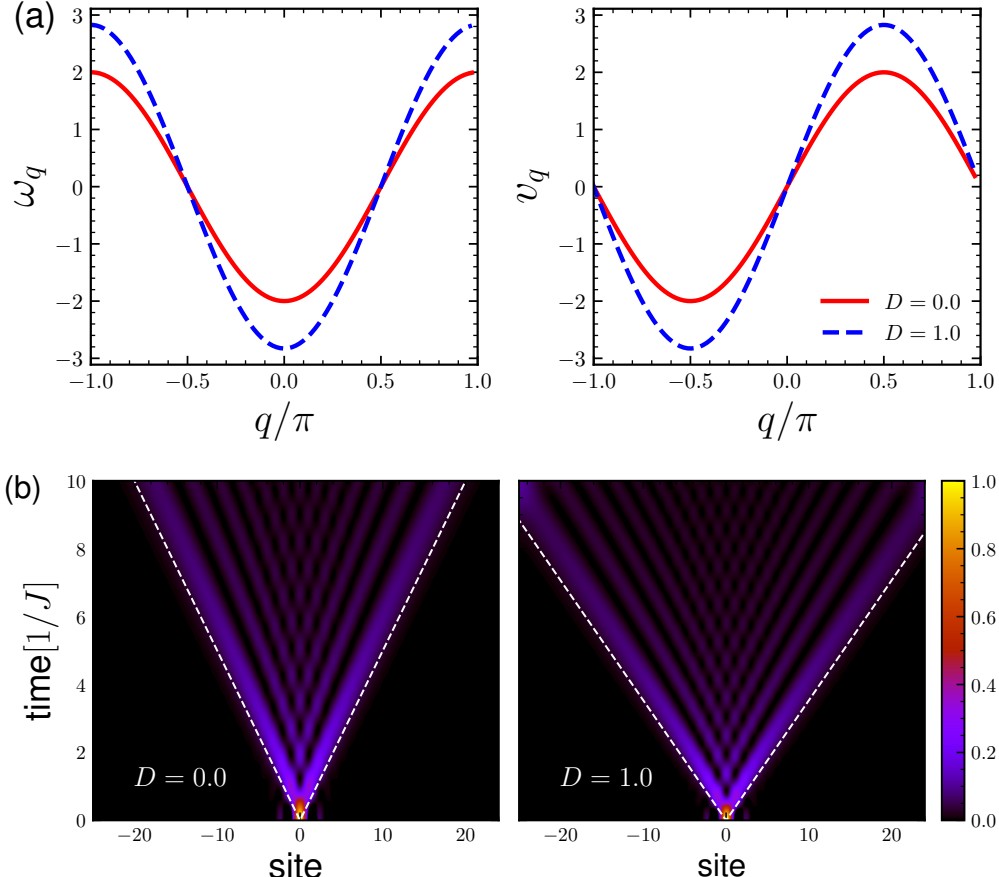

Figure 2: (colour online) (a) Energy dispersion (left panel) relations for Hamiltonian in Eq.(4) for $D = 0.0$ and $D = 1.0$. The corresponding group velocity $v_g = \frac{d\omega_q}{dq}$ are shown in the right panel. (b) single site occupation $\langle \psi_t | c_j^\dagger c_j | \psi_t \rangle$ versus time. Excitation at $L/2$ spreads light-cone-like, bonded by the maximal group velocity $v_{\max} = 2\tilde{J}$, shown with dashed line.

where $\tilde{J} = \sqrt{J^2 + D^2}$. Note that we performed a unitary transformation $H_{\text{fermion}} = \mathcal{Q} H \mathcal{Q}'$ to eliminate the $D$ term. Where $\mathcal{Q} = \prod_{j \in even} e^{-i\theta\sigma_j^z}$, and $\tan(\theta) = -D/J$. This model has been well studied in the literatures [32–34]. Rewriting the local spin-lowering and spin-raising operators $\sigma_j^\pm = (\sigma_j^x \pm i\sigma_j^y)/2$, then with a Jordan-Wigner transformation, these operators can be mapped to anti-commuting quasiparticles via $c_i = \prod_{j<i}(-1)^{\sigma_j^+ \sigma_j^-} \sigma_i^- = \prod_{j<i}(1 - 2\sigma_j^+ \sigma_j^-)\sigma_i^-$. This end up with a one-dimensional noninteracting spinless fermion Hamiltonian: $\mathcal{H}_{\text{fermion}} = \sum_j \tilde{J}(c_j^\dagger c_{j+1} + \text{h.c.}) + hc_j^\dagger c_j$. By doing a Fourier transformation into the momentum space as $c_j = L^{-1/2} \sum_q e^{-iqj} c_q$, one get diagonalised Hamiltonian as $\mathcal{H}_{\text{fermion}} = \sum_q \omega_q c_q^\dagger c_q$, where $\omega_q = 2(\tilde{J} \cos q + h)$. For $L$ spins, the quasi-momenta are given by $q = n2\pi/L$, where $n = -L/2, \ldots, L/2 - 1$. The group velocity of quasiparticles is given by $v_g = \frac{d\omega_q}{dq}$. Fig.2-(a) shows energy dispersion relation $\omega_q$ and the corresponding group velocity. It can be noticed that the energy is bounded with band width $4\tilde{J}$, and the DM interaction normalizes the coupling with a visible impact on maximum group velocity $v_g^{\max} = 2\tilde{J}$ at $q = \pm\pi/2$.

Now lets explore the out-of-equilibrium dynamics. We prepare an initial state $|\psi_0\rangle = c_{L/2}^\dagger |0\rangle$, which in fermionic language means crating a single quasiparticle at the middle of chain as $|0 \cdots 00100 \cdots 0\rangle_L$. This fermionic state can be easily prepared in experi-

ments [35]. Then we probe spreading of excitation in system with monitoring single site occupation $\langle \psi_t | c_j^\dagger c_j | \psi_t \rangle$, where $|\psi_t\rangle = L^{-1/2} \sum_q e^{i(qL/2 - \omega_q t)} c_q^\dagger |0\rangle$. With a little algebra we have $\langle c_j^\dagger c_j \rangle_t = L^{-2} \sum_{q_1, q_2} e^{-i(q_1 - q_2)(L/2 - j)} e^{i(\omega_{q_2} - \omega_{q_1})t}$. Note that connection with the original spin model can be traced with $\langle \sigma_j^z \rangle_t = 2\langle c_j^\dagger c_j \rangle_t - 1$, which means, in Fig.2-(b), we are looking at the correct observable.

Fig.2-(b) shows results for a chain with $L = 51$ sites. A locally perturbed system causes emitting quasiparticles at different speeds, which the fastest particles propagate at a speed $v_g^{\max} = 2\tilde{J}$. It gives rise to LR bound, which defines an effective causal cone for spatial correlations, outside of which the correlations are exponentially suppressed [6]. It is readily apparent that with the perturbing system at the center, excitations propagate in a symmetric way to both sides and a perfect light cone is constructed. As discussed, tuning DM interaction increases $v_g^{\max}$, and this is clear by comparing left ($D = 0.0$) and right ($D = 1.0$) panels in Fig.2-(b). For the time window present in these figures, quasiparticles for the case $D = 1.0$ hit the boundaries. Fig.2-(a) shows that for regime $\alpha \to \infty$, the dispersion and corresponding velocity are bounded. This leads to a well-defined boundary of light-cone, which is clear in Fig.2-(b).

## 3.2  Mapping to bosonic particles ($\Delta = 0.0$)

The presence of long-range interactions makes it impossible, as the Jordan-Wigner transformation is a non-local string operator, to map directly from the spin system to fermionic particles. However, we can use linear spin-wave (LSW) theory to describe Hamiltonian in terms of quantum fluctuations around its classical ground state. For model Hamiltonian Eq.(1) with the absence of DM interaction, the validity of LSW in the presence of power-law interactions has recently been addressed [36]. Moreover, for the initial state considered here, as is composed of a single magnon, the LSW treatment is exact.

We can map spin particles onto a system of hard-core bosons (magnons), introducing the Holstein-Primakoff transformation, $\sigma_j^z \to n_j - 1/2$, $\sigma_j^+ \to b_j^\dagger$, and $\sigma_j^- \to b_j$ where $b_j^\dagger (b_j)$ are the creation (annihilation) operators of hard-core bosons and $n_i$ is the number operator at site $i$. These bosons obey the bosonic commutation relationships, $[b_j, b_j^\dagger] = 1$, with constrain $b_j^2 = (b_j^\dagger)^2 = 0$, such that a site either filled by one boson or empty. This can be interpreted, a spin up particle is represented by a filled site and a spin down particle by an empty site. To treat the dynamic, the picture of non-interacting magnons is only valid when $L^{-1} \sum_q \langle b_q^\dagger b_q \rangle(t) \ll 1$ at all times, otherwise one has to account for the presence of magnons interactions. Considering the initial ordering of the spins along the $z$-axis (as close to initial state prepare before quench in this work) shown be a good approximation for $\Delta = 0$ [9, 36].

Then we end up with a noninteracting magnonic (bosonic) Hamiltonian

$$\mathcal{H}_{\text{boson}} = \sum_{ij} \left( \mathcal{J}_{ij} b_i^\dagger b_j + \text{h.c.} \right) + h \sum_j b_j^\dagger b_j, \tag{5}$$

where $\mathcal{J}_{ij} = J_{ij} + iD_{ij}$. The complex hopping, can be recast into a phase like term as $\mathcal{J}_{ij} = \frac{\tilde{J} e^{i\phi}}{|i-j|^\alpha}$ with $\phi = D/J$ independent of the decay exponent $\alpha$. By doing a Fourier transformation into the momentum space as $b_j = L^{-1/2} \sum_q e^{-iqj} b_q$, one get diagonalised Hamiltonian as $\mathcal{H}_{\text{boson}} = \sum_q \epsilon_q b_q^\dagger b_q$, where $\epsilon_q = \tilde{J} \sum_{r \neq 0} \cos(qr - \phi) r^{-\alpha} + h$. In contrast to the nearest-neighbour limit, for long-range interaction on a finite system with open boundary conditions, maximal group velocity can be extracted with $v_g^{\max} \equiv |\max_q \left( \epsilon_{q+\pi/(L+1)} - \epsilon_q \right)(L+1)/\pi|$. In contrast to the nearest-neighbor case ($\alpha \to \infty$), now the DM interaction is not gauging away with helping the unitary transformation $\mathcal{Q}$.

Figs.3-(a,b) show the energy dispersion relation $\epsilon_q$ and the corresponding group velocity for various interaction ranges. In the absence of DM interaction ($D = 0$, left panels), a clear

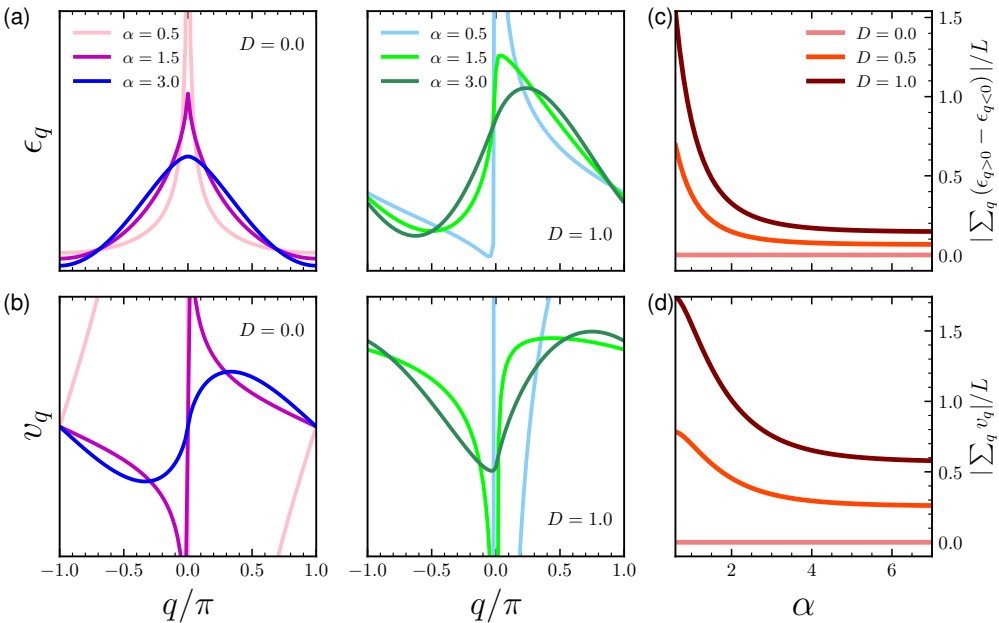

Figure 3: (color online) (a,b) Energy dispersion $\epsilon_q$ and corresponding group velocity $v_g = \frac{d\epsilon_q}{dq}$ of Eq.(5) for various interactions range. Results of two DM interactions shown on left ($D = 0.0$) and middle ($D = 1.0$) panels, respectively. (c,d) A figure of merit which qualitatively shows asymmetric features of energy and group velocity.

difference going from short-range regime to long-range regime is visible. For regime $\alpha > 2$, the energy dispersion as well as its derivative are bounded i.e. $v_g^{\max} \equiv \epsilon_q' < \infty$. As we already noted, this ends in a clear light-cone shape of correlation spread. The situation changes for $\alpha < 2$ as a kink appear at zero momentum $q = 0$. For the regime where $1 < \alpha$, although the dispersion is bonded $\epsilon_q < \infty$, the velocity diverges. This enhances leakage of correlations outside of the light cone. On the other hand when $\alpha < 1$, both dispersion and group velocity become unbounded, and the light-cone disappears.

Now let us turning on the DM interaction. The dispersion relation and corresponding group velocity for $D = 1.0$ are illustrated on the middle panels of Figs.3-(a,b). As before, by changing interaction from short to long-range, a kink appears at $q = 0$. However, an asymmetry is noticeable in energy dispersion and also group velocity, which develops by complex hopping accompanied by phase shift in the model. As emerges, the asymmetry is getting substantial if the model exists at a long-range regime. To attain a deeper understanding, in Figs.3-(c,d), we plot a figure of merit to depict qualitatively asymmetry dependence versus decay exponent $\alpha$. As demonstrated, the figure of merit is zeros when $D = 0.0$, whereas for $D \neq 0$ model has a certain level of asymmetry at short-range and gradually increases towards the long-range regime ($\alpha \to 0$). Analytically, one can also show that the presence of the DM term accompanied with interaction range, $\alpha$, leads to asymmetric excitation propagation throughout the model. To this end, let us assume $\phi < 1$ and expand velocity about momentum $q = 0$. Then by considering contribution of the lowest modes to the velocity gives $v_{q=0^+} - v_{q=0^-} \approx \frac{4\pi\phi}{L} r^{1-\alpha}$. This is in agreement with the figure of merit shown in Fig.3-(d).

This feature greatly impacts on correlation spreading of the model. We will show it within the quench setup introduced in the proceeding part.

We prepare an initial state with one boson at the center of the chain as $|\psi_0\rangle = b_{L/2}^\dagger |0\rangle$, which this bosonic state also can be prepared in experiments [37]. Fig.4 displays spatially resolved single site occupation $\langle\psi_t| b_j^\dagger b_j |\psi_t\rangle$ versus time for $\alpha = 1.1$. Note that connection

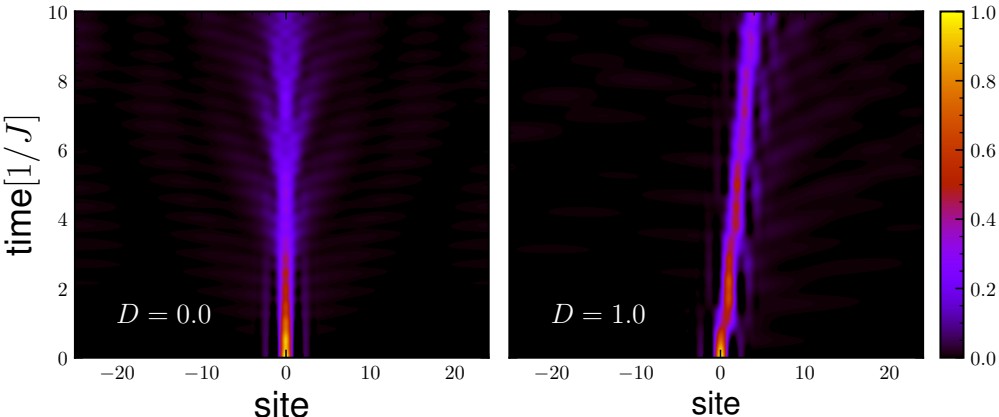

Figure 4: (colour online) Single site bosonic particle occupation $\langle \psi_t | b_j^\dagger b_j | \psi_t \rangle$ versus time for $D = 0.0$ and $D = 1.0$ depicted on left and right panels, respectively. Parameter $\alpha = 1.1$.

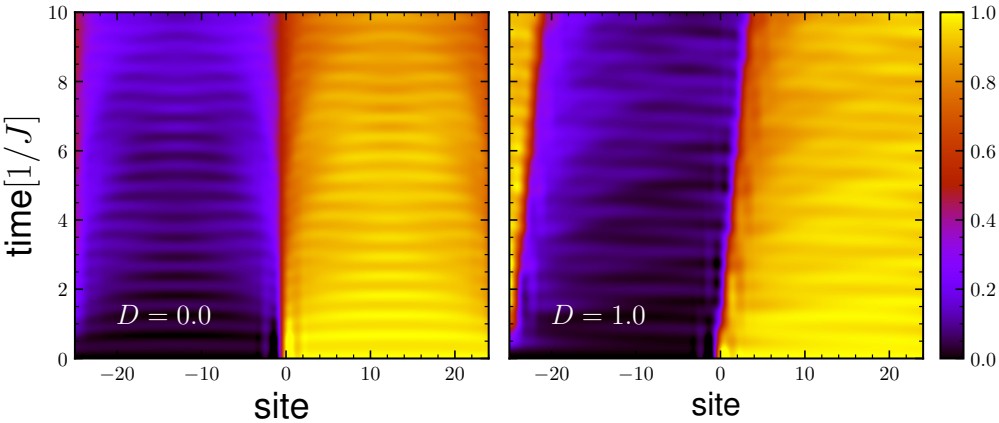

Figure 5: (colour online) Same as Fig. 4, but for a domain-wall state as spread over $x > 0$ as $|\psi_0\rangle = c_{L/2}^\dagger \cdots c_L^\dagger |0\rangle$.

with original spin model reads as $\langle \sigma_j^z \rangle_t = \langle b_j^\dagger b_j \rangle_t - 1/2$. In the absence of DM interaction, excitation at the center propagates to the both sides of the chain symmetrically. Although the wavefront needs finite time, the light cone is not as clear as the short interaction regime [7]. The situation changes when DM interaction is turned on. It is apparent that wavefront propagation is no longer symmetric. Indeed, if we look more closely at the right panel of Fig.4, we see that long-range DM interaction behaves as a barrier and guides the spin-wave into the desired direction [38, 39]. By changing the DM interaction, namely from $\hat{z}$ to $-\hat{z}$, one can reverse propagation direction. This could be understood as follows: the phase in the hopping (in hard-core boson language) does not do anything if one has nearest-neighbor interactions, however at long-range interactions, one has closed loops, and now there are interference effects between different hopping paths where the phases come into play. This can also be rephrased in terms of the unitary transformation, which does not amount to a simple shift of the dispersion relation when interactions are longer ranged because the lattice is no longer bipartite.

Before closing this part, it is worth commenting on that the magnon propagation with a dispersion $\epsilon_q \approx \cos(qr - \phi)$ is not parity symmetric. With choosing an initial state which is

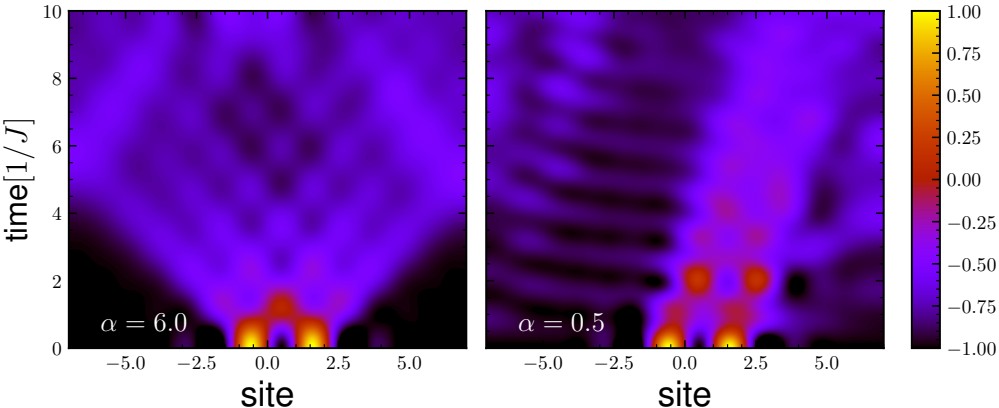

Figure 6: (color online) Numerical exact results of spatially and temporally resolved spin polarisation $\langle \sigma_i^z \rangle_t$ of XXZ model (see Eq.(1)) with $\Delta = 1.5$ for $D = 1.0$ at short ($\alpha = 6.0$, left panel) and long ($\alpha = 0.5$, right panel) range interacting regimes.

perfectly localized in space (or a uniform superposition of in momentum space), and since the dispersion relation is just shifted in momentum space, it will continue to look symmetric. However, considering a different initial state, e.g. a magnon wave-packet spread over a width of $x > 0$ lattice sites, this would be nonuniform in momentum space, preferentially selecting momenta around zero. Thus, this wave-packet would move asymmetrically due to DM interactions. To verify this, we consider a domain-wall state $|\psi_0\rangle = c_{L/2}^\dagger \cdots c_L^\dagger |0\rangle$. As can be seen in Fig.5, while for the case with $D = 0.0$ wave packet almost confined into $x > 0$, for the case with $D \neq 0$ nonuniform momentum contribution combined with phase shift modulated with DM interaction leads to asymmetry propagating of excitation through the model.

### 3.3 Exact diagonalization

Testing the preceding results in a many-body picture demands numerical techniques to consider Eq. (1), as it is not trivial to tackle analytically. Using the Krylov-space technique, we probe the spin polarisation $\langle \sigma_i^z \rangle_t$. For an initial state, without loss of generality, we choose a two-body state as $|\psi_0\rangle = c_{L/2-1}^\dagger c_{L/2+1}^\dagger |0\rangle$. Fig.6 depicts results for the XXZ model with $\Delta = 1.5$. Consistent with the fermionic picture, at short-range interaction, the initially localized perturbation spread bounded by LR velocity, leads to the formation of the light cone (see the left panel of Fig.6). Although, the presence of DM interaction changes the propagation speed and induces a degree of asymmetry, the light cone still exists (as the Hamiltonian is local and LR bound is well defined). Increasing interaction range (decreasing $\alpha$) leads the correlations instantaneously spread over the chain as shown in the right panel of Fig.6. This confirms the breakdown of LR bound similar to that of Ref. [3, 4, 7].

By turning the DM interaction, we find results similar to the single-particle case (see the right panel of Fig.6). Interestingly, with complex interference effects induced by DM coupling, spin excitation guides to the desired direction. This calls that one could expect these behaviors to survive even in the many-body regime, as recently an asymmetric transport also reported in Abelian anyons [28]. However, contrary to the Abelian anyons in which non-local anyonic commutation plays a role, for the model proposed in this work complex quantum interference and non-trivial shift of dispersion relation (as the lattice is no longer bipartite with long-range interaction) are possible scenarios to explain asymmetric transport.

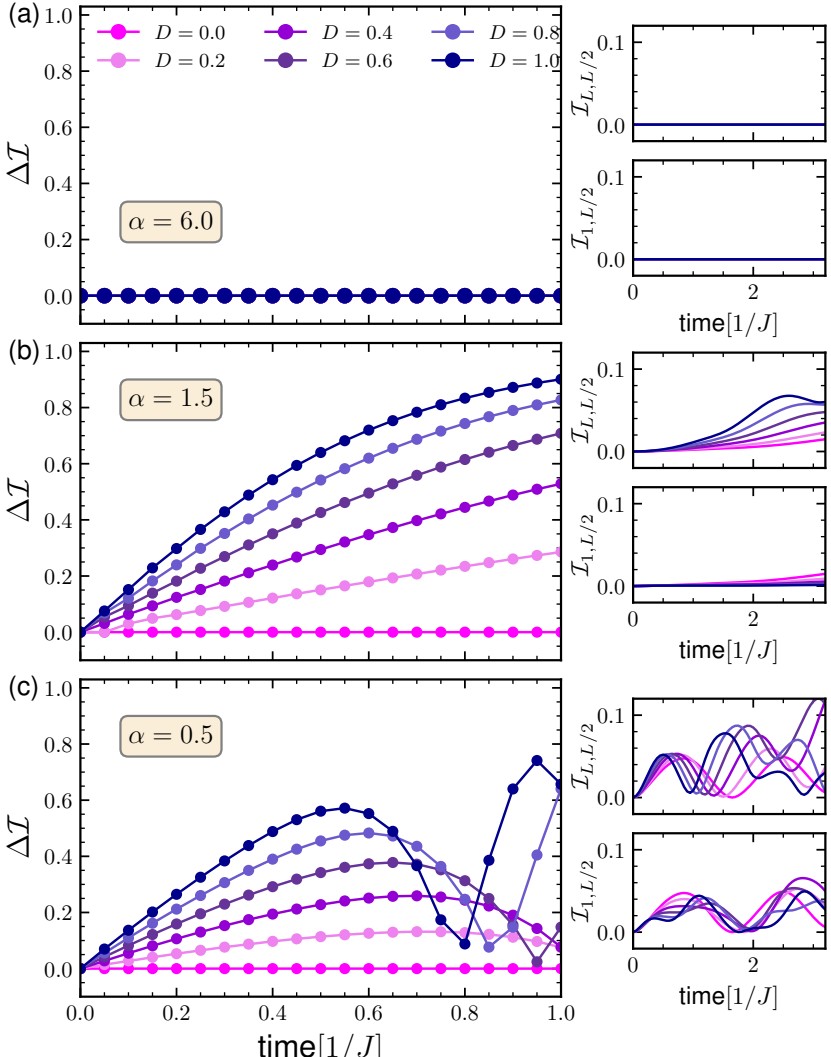

Figure 7: (colour online) Main panels (a-c) show time evolution of quantum mutual information polarization $\Delta\mathcal{I}$ between spin at centre (place of initial perturbation) and the spins located at edges of chain. Side panels correspond to the quantum mutual information $\mathcal{I}_{ij}$ of the main panel. Data for chain size $L = 23$ and $\Delta = 0.5$.

## 3.4 Quantum information tools

It would also be interesting to look at the entanglement spreading in the setup. To this end, we first numerically analyze block entanglement entropy.

Fig.8 displays growth of half-chain entanglement entropy $S_{\text{vN}}$. In all sets of parameters shown here, $S_{\text{vN}}(t)$ initially grows as a power of time $t$ and at longer time saturates to $S_{\text{vN}}(t) = \log 2$ independent of system regime. Figs.8(a,b) present results for long-range ($\alpha = 0.5$) and short-range ($\alpha = 6.0$) regimes with different DM coupling strengths. By increasing DM, the linear(in logarithmic scale) behavior of $S_{\text{vN}}$ breaks down and changes into an oscillatory before entering the saturation regime. As observed, while the presence of DM coupling modulates a higher value of entanglement at a smaller time, it finds lower entanglement at later times. This may be explained in connection to the confinement of elementary excitations [13, 40]. It leads to the suppression of entanglement entropy which in general instances turns out to oscillate rather than grow indefinitely.

The situation is more profound in the long-range regime as correspondingly excitations speed modified by DM interaction. Figs.8(c,d) illustrate results for a fixed DM coupling by changing the power exponent $\alpha$. In absence of the DM coupling (Fig.8-(c)) and for $\alpha > 1.0$, the half-chain entropy initially increases as a power of $t$, while it starts to oscillate at short times for the long-range regime. Remarkably, this becomes more prominent with the presence of DM interaction. In Fig.8(d), it can be seen that by changing $\alpha$, $S_{\mathrm{vN}}$ shows appreciable oscillation with a few orders of magnitude reduction. This may show a signature of diffusive rather than ballistic transport in the chain. Although a detailed study in this line needs separate work, we give some elaboration on this issue in the appendix. Another explanation would be that the dynamics are constrained to take place in a small part of the total available Hilbert space. This is already addressed in Ref. [8], in the case of infinite-range interactions with connection to the Lipkin-Meshkov-Glick Hamiltonian [41, 42].

At the experimental level, this is more straightforward to measure quantum mutual information than the von Neumann entropy. So further confirmation may be accessed by inspection quantum mutual information between two distant spins $\mathcal{I}_{ij}$. In Fig.7, we have plotted a figure of merit to measure quantum information polarization

$$\Delta \mathcal{I} = \left| \frac{\mathcal{I}_{1,L/2} - \mathcal{I}_{L,L/2}}{\mathcal{I}_{1,L/2} + \mathcal{I}_{L,L/2}} \right| ,$$

between spin located at the center with spins is sitting at the edges of the chain. For comparison purposes, results for three different regimes are presented in the same time window.

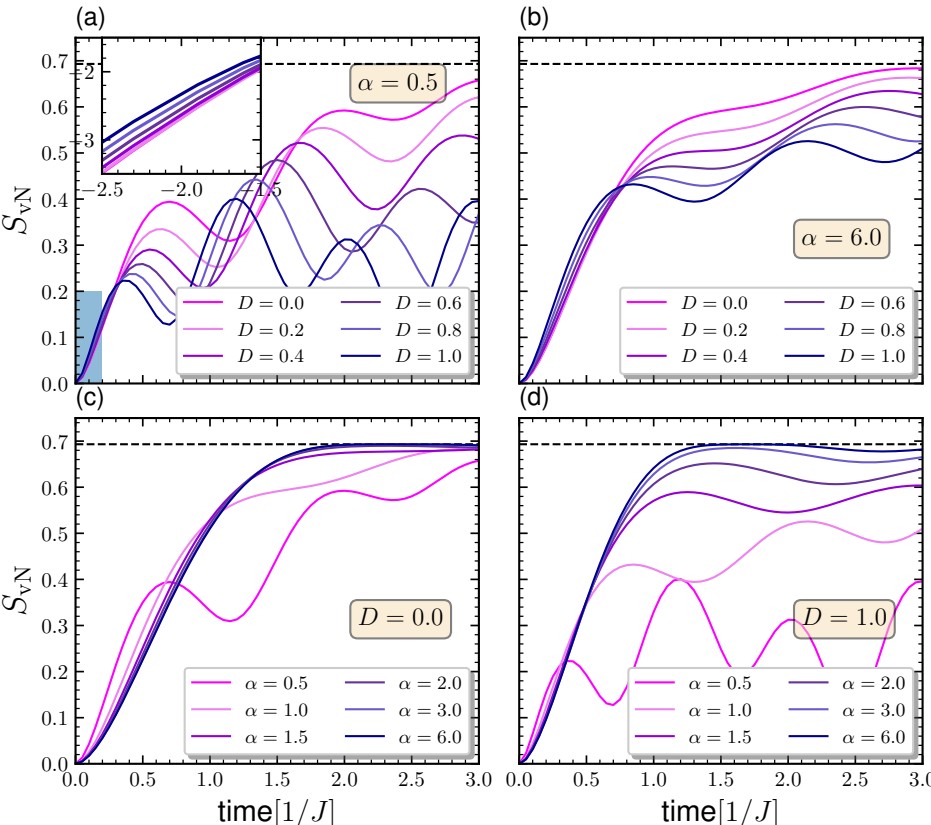

Figure 8: (colour online) Growth of half-chain entanglement entropy $S_{\mathrm{vN}}$ for various parameters in the Hamiltonian of Eq.(1). The horizontal dashed line is saturate value $S_{\mathrm{vN}}(t) = \log 2$. Insert plot illustrates shaded area in logarithmic scale. Data for chain size $L = 23$ and $\Delta = 0.5$.

Noticeably, we find that for the intermediate and the long-range regime distant spins become entangled instantaneously, whereas for the short-range regime quasiparticle needs a certain time to reach edges (consistent with LR bound). Furthermore, we notice that for $D = 0$ and irrespective the interaction range, $\Delta\mathcal{I}$ is always zero. This confirms the symmetric spread of quantum information to either sides of the chain. However, for a finite $\alpha$ (appreciable for $\alpha \lesssim 2$), the presence of DM interaction changes the way quantum information spreads over the chain. Interestingly, as shown in Fig.7, by gradually tuning the DM coupling strength quantum information $\Delta\mathcal{I}$ increases and even may reach a fully polarized one for some time intervals.

## 4 Summary and Outlook

Recently, long-range spin chains have gained attention as platforms to study quantum information dynamics. One can use the possibility of controlling the power-law interactions and vector chirality of DM coupling in order to drive information along the chain. We proposed a protocol setup consisting of long-range spin-1/2 chain modulated with DM coupling. For translationally invariant fermionic or bosonic models, information spreading occurs in a spatially symmetric way. However, our results reveal that by adjusting the interaction range and DM coupling direction transport excitation can be guided in desired path. This could potentially enable us to design quantum information protocols with unidirectional and bidirectional quantum channels, and maybe testable with current state-of-the-art trapped-ion experiments. We further explore the growth of block entanglement entropy in these systems and an order of magnitude reduction distinguished. This could also be interesting to simulate the quantum system on a computer, as linear growth of entanglement with time, often makes numerical simulation unfeasible. [43–45]. Finally, it would be interesting to study the effects of long-range DM interactions on other systems, e.g. many-body localized systems [46, 47], or long-range disorder spin chains [48].

## Acknowledgement

I would like to express my gratitude to Prof. Christoph Karrasch for the financial support of my stay in his group. I acknowledge the support from Deutsche Forschungsgemeinschaft (DFG) KE-807/22-1. Thanks to Hasan Fathi and Youcef Mohdeb for proofreading the article. The ED calculations have been performed at the Centre De Calcul (CDC), CY Cergy Paris Université. We also thank the anonymous referees for the thoughtful comments that resulted in the improvement of the paper.

## A    Half-chain entanglement growth

In this appendix, we aim to have more analysis on $S_{vN}$. Fig.9 plots results of entanglement growth for different system sizes. Interestingly, the results at the long-range regime shows clear size dependence with reduction of entanglement growth at thermodynamic limit. The DM interaction effects are also visible with comparing panels (a) and (b). Although, at the long-range regime and for short times, $S_{vN}$ increases faster than the nearest- neighbor case (see the solid-black line in Fig.9), but a suppression at a later time is profound

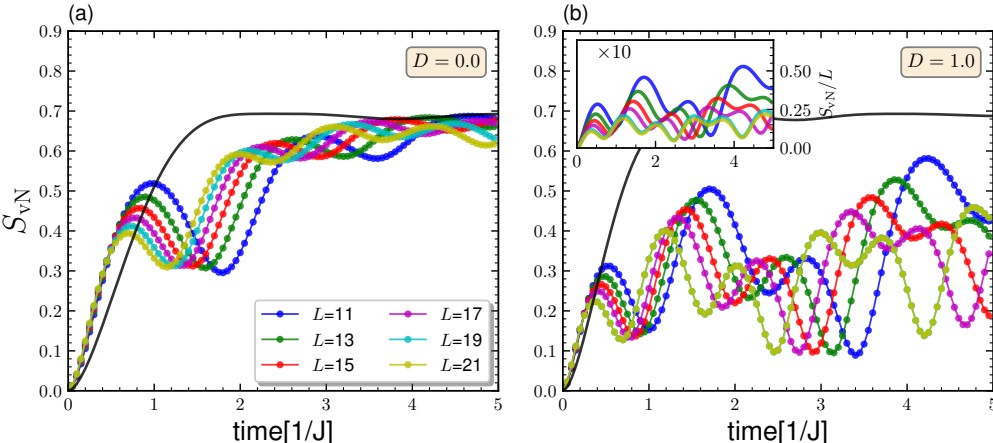

Figure 9: (colour online) Growth of half-chain entanglement entropy $S_{vN}$ of Hamiltonian Eq.(1) with $\Delta = 0.5$ for various different chain length at long-range regime with $\alpha = 0.5$. Where DM interaction (a) $D = 0.0$, (b) $D = 1.0$. The solid-black line is data for nearest-neighbour case ($\alpha \rightarrow \infty$). Inset shows $S_{vN}$ scaled with chain length.

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
