# Peer review of "Asymmetric Transport in Long-Range Interacting Chiral Spin Chains"

_SciPost Physics, doi:SciPost Phys. Core 5, 021 (2022)_

## Round 1 · Referee Report · Anonymous (Referee 1) · 2021-9-16

Report

I have reviewed the paper entitled “Asymmetric transport in long-range interacting chiral spin chains”, by Javad Vahedi. The paper investigates the effect of Dzyaloshinskii-Moriya (DM) interactions in one-dimensional spin chains, with power-law decaying interactions, onto the spreading of correlations after a local quench. In principle, the problem is interesting and relevant to the readership of Scipost, but I am not sure that all presented results are valid, and there are major inconsistencies in the paper. Therefore, I do not think the paper should be published, at least not without major revisions.

1) First of all, the paper is poorly written and contains many typos and grammatical errors. Before any resubmission, I strongly suggest the author to ask a fluent english-speaking collegue to read the paper carefully.

2) When mapping the spin chain onto free fermions (III.A), the quantities of interest are still the spin correlations. Instead, the fermion correlations are presented in Fig. 2. The author should instead consider the (initial, i.e. unrotated) spin variables, and study the evolution of the mean magnetization, and possibly also the spin-spin correlations.

3) In III.B, the use of LSW approximation should be further discussed. Around which mean-field state is it performed? Is the criterion $\langle b_i^\dagger b_i \rangle \ll ½$ valid at all sites and at which times? Furthermore, here also, one is primarily interested in the spin variables, and not in the bosonic variables (see Fig. 4). The author should discuss the relation between them.

4) In the caption of Fig. 5, the Ising model cannot be simply $\Delta → \infty$. A rescaling has been made, and should be explicited by the author.

5) The different regimes of interaction range are not convincing. For instance, one clearly sees the asymetry of the dispersion relation for all $\alpha$ when $D > 0$. Why would it have no consequence on the spreading of correlations? Furthermore, the asymmetry of the mutual information (III.D, Fig. 7), is clearly present for all $\alpha$. This is in contradiction with the main claims of the paper.

6) The author finds that the half-chain entropy is reduced at long times in the presence of DM interactions. Why is it so? Does the author have a physical explanation?

7) In the appendix, the author investigates the level statistics of the models, but this is not discussed in the main text. What is the added value of this study?

In conclusion, although the topic is a priori interesting, I have many doubts on both the technical validity of the results, and on their consistent interpretation. Therefore, I cannot recommend the publication without major revisions, and an almost complete rewriting of the paper.

  • validity: -
  • significance: -
  • originality: -
  • clarity: -
  • formatting: -
  • grammar: -

Author:  Javad Vahedi  on 2022-02-09  [id 2179]

(in reply to Report 1 on 2021-09-16)
Category:
reply to objection

We thank the anonymous referee for reading the paper carefully and providing thoughtful comments, many of which have resulted in changes to the revised version of the manuscript. Below you can find, reply to the comments.

Report 1: I have reviewed the paper entitled “Asymmetric transport in long-range interacting chiral spin chains”, by Javad Vahedi. The paper investigates the effect of Dzyaloshinskii-Moriya (DM) interactions in one-dimensional spin chains, with power-law decaying interactions, onto the spreading of correlations after a local quench. In principle, the problem is interesting and relevant to the readership of Scipost, but I am not sure that all presented results are valid, and there are major inconsistencies in the paper. Therefore, I do not think the paper should be published, at least not without major revisions.

1) First of all, the paper is poorly written and contains many typos and grammatical errors. Before any resubmission, I strongly suggest the author to ask a fluent english-speaking collegue to read the paper carefully.

Thanks the referee for this comment. We try to polish and resolve typos in the text.

2) When mapping the spin chain onto free fermions (III.A), the quantities of interest are still the spin correlations. Instead, the fermion correlations are presented in Fig. 2. The author should instead consider the (initial, i.e. unrotated) spin variables, and study the evolution of the mean magnetization, and possibly also the spin-spin correlations.

Thanks the referee for this comment. We added We correct it. Please see lines 218-221 and 332-334

3) In III.B, the use of LSW approximation should be further discussed. Around which mean-field state is it performed? Is the criterion valid at all sites and at which times? Furthermore, here also, one is primarily interested in the spin variables, and not in the bosonic variables (see Fig. 4). The author should discuss the relation between them.

Thanks the referee for this comment. We add some explanation in this regard. The relation between and boson is also mentioned. Please see lines: 248-250 and 264-268

4) In the caption of Fig. 5, the Ising model cannot be simply . A rescaling has been made, and should be explicited by the author.

Thanks the referee for this comment. We change section 4 in the revised version and correspondingly Fig.5 is also changed.

5) The different regimes of interaction range are not convincing. For instance, one clearly sees the asymmetry of the dispersion relation for all when . Why would it have no consequence on the spreading of correlations? Furthermore, the asymmetry of the mutual information (III.D, Fig. 7), is clearly present for all . This is in contradiction with the main claims of the paper.

Many thanks for the comment. That is totally true, asymmetry exits for all $\alpha$ as soon as $D\neq 0$. This whys you can see the asymmetry in III.D, Fig. 7. But for a fixed $D$, say $D=1.0$ we have considered in the paper, a really long-range $\alpha$ needs to see an appreciable effect on spreading.

6) The author finds that the half-chain entropy is reduced at long times in the presence of DM interactions. Why is it so? Does the author have a physical explanation?

Many thanks for the comment. We add some explanation in this regard. Please see line: 424-424 and 443-448.

7) In the appendix, the author investigates the level statistics of the models, but this is not discussed in the main text. What is the added value of this study?

Thanks for this comment. We have removed this part totally from this draft. We will consider it for a related project later.

In conclusion, although the topic is a priori interesting, I have many doubts on both the technical validity of the results, and on their consistent interpretation. Therefore, I cannot recommend the publication without major revisions, and an almost complete rewriting of the paper.

---

## Round 1 · Referee Report · Anonymous (Referee 2) · 2021-10-18

Report

My overall assessment of the paper is positive, as the author has considered a truly interesting, theoretically well motivated and experimentally relevant problem, which to the best of my knowledge has not been addressed before in this form. The presentation is sound and previous literature is mostly reported and correctly cited. Above all, to further credit of the author, I believe the presented main result - i.e. the discovery of asymmetric particle transport due to the combination of long-range and DM interactions - is correct, although the interpretation of some numerical and analytical results seems incorrect or rather incomplete (see below), such that the whole presentation sounds a bit misleading. For these reasons, I recommend publication of this work in SciPost Physics only after the author carefully amends the various issues I discuss below. Note that some of them address points also raised by a previous Referee in Report 1.

1) As in Report 1, I also recommend revision of the grammar and misprints in the text 2) The author writes two times that the light-cone ceases to exist only for alpha<1 (intro and page 4). This is misleading, as the light-cone effect does not exist as soon as the maximum propagation velocity becomes infinite, i.e., for alpha<2 (as the author recognizes). I recommend to modify the phrasing 3) page 2 second column: Ref 7 is cited as an experiment, while it is a numerical simulation 4) the second appendix reports the study of level statistics. While its utility is questionable, as the author always works in the single-particle sector (see below), the reported results suggest integrability of the nearest-neighbor interacting XXZ+DM spin chain. The author’s comment on this is not clear enough, as it should refer to the staggered unitary rotation Q performed on page 3, which maps the model to a standard XXZ chain. 5) Sec IIIA: As also commented in Report 1, the author should be interested in the density in the original spin variables, whereas the text refers to the transformed fermion density. The author should comment that this is unchanged under the transformation Q, so in practice we are looking at the correct observable. 6) Sec IIIA: the author reports symmetric propagation. I believe this result is correct, however: 1) Fig 2b is for some reason asymmetric, which makes it unnecessarily hard for the reader to grasp the simple result 2) the presentation obscures the main understanding of this result: when viewed in the original fermionic variables (before the transformation Q), the magnon propagates with a dispersion relation e_k = \tilde{J} \cos(k-\phi). Thus, propagation is not parity symmetric. The only reason why it looks symmetric, is because the initial state is perfectly localized in space, and thus it is a uniform superposition of all momenta. Since the dispersion relation is just shifted in momentum space, it will continue to look symmetric. However, had the author considered a different meaningful initial state, e.g. a magnon wavepacket spread over a width of x>1 lattice sites, this would be nonuniform in momentum space, preferentially selecting momenta around zero. Thus, this wavepacket would move asymmetrically due to DM interactions. I suggest the author to add this explanation and include a simulation showing this effect. 7) Sec IIIB: The presentation of linear spin wave theory is misleading. In fact, the LSW treatment is exact in the case presented by the author, because the initial state is composed of a single magnon. What’s more, there is no difference between Jordan-Wigner fermions and hardcore bosons (magnons) in this sector. 8) Sec IIIB: The result that long-range interactions produce asymmetric transport even when short-range interactions don’t, seems to be correct. However, the author should improve the analytical understanding of this fact, building on the derived dispersion relation. For example, it is possible to determine the difference between left and right maximum velocities analytically as a function of the parameters, including alpha. I believe that the asymmetry should be present for all alpha’s, but is only appreciable in the plots for small alpha. Fig 3 middle panel suggests this, and the right panel attempts to quantify this, but - I believe - in an incorrect way. In conclusion, if I am right, the description of the main result of the paper should be modified accordingly. 9) Sec IIIB: similarly to a previous comment, cf also Report 1, the author should comment that the magnon density is the same as the spin density one is interested in in the original formulation of the problem. Also, above Fig 4 there is a typo: it should be b^dagger b rather than c^\dagger c. 10) Sec IIIB: the explanation at the end seems correct and insightful. One can also phrase it in terms of the staggered transformation Q, which does not amount to a simple shift of the dispersion relation when interactions are longer ranged because the lattice is no longer bipartite. 11) Sec IIIC: I believe this is the weakest point of the paper. As results from the last paragraph of the section, unfortunately the author doesn’t seem to have realized that as long as one considers an initial state in the single particle sector, the interaction \Delta is completely immaterial (there is no other particle with which to interact). Indeed, the three plots in Fig 5 look exactly the same despite the widely different values of the interaction. To study the effect of interactions, the author should consider initial state with multiple particles, such as those considered in Ref. 25. I suggest the author to do this, or to remove Fig 5 and the discussion of interactions altogether from the paper to avoid any misinterpretations by the readers. If this is not correct, the author should anyway discuss this point to avoid confusion. 12) Sec IIID: apart from Refs. 7,8,9,11,12, the author should cite Phys. Rev. Research 2, 012041(R) for the effect of long-range interactions on the entanglement dynamics. Note that following comment 11, the dynamics of entanglement would become truly more interesting in the manybody sector, as entanglement entropy would be created not only by the uncertainty in the position of the particle across the cut (and hence bounded by log2 as in Fig 8) but also by genuine quantum correlations shared by the particles.

  • validity: good
  • significance: good
  • originality: high
  • clarity: good
  • formatting: good
  • grammar: below threshold

Author:  Javad Vahedi  on 2022-02-09  [id 2180]

(in reply to Report 2 on 2021-10-18)
Category:
reply to objection

We thank the anonymous referee for reading the paper carefully and providing thoughtful comments, many of which have resulted in changes to the revised version of the manuscript. Below you can find, reply to the comments.

Report 2: My overall assessment of the paper is positive, as the author has considered a truly interesting, theoretically well motivated and experimentally relevant problem, which to the best of my knowledge has not been addressed before in this form. The presentation is sound and previous literature is mostly reported and correctly cited. Above all, to further credit of the author, I believe the presented main result - i.e. the discovery of asymmetric particle transport due to the combination of long-range and DM interactions - is correct, although the interpretation of some numerical and analytical results seems incorrect or rather incomplete (see below), such that the whole presentation sounds a bit misleading. For these reasons, I recommend publication of this work in SciPost Physics only after the author carefully amends the various issues I discuss below. Note that some of them address points also raised by a previous Referee in Report 1.

As in Report 1, I also recommend revision of the grammar and misprints in the text.

Thanks the referee for this comment. We have tried to polish the text and resolve typos in the text.

2) The author writes two times that the light-cone ceases to exist only for alpha<1 (intro and page 4). This is misleading, as the light-cone effect does not exist as soon as the maximum propagation velocity becomes infinite, i.e., for alpha<2 (as the author recognizes). I recommend to modify the phrasing.

Thanks the referee for this comment. We correct it.

3) page 2 second column: Ref 7 is cited as an experiment, while it is a numerical simulation.

Thanks the referee for this comment. We correct it.

4) the second appendix reports the study of level statistics. While its utility is questionable, as the author always works in the single-particle sector (see below), the reported results suggest integrability of the nearest-neighbor interacting XXZ+DM spin chain. The author’s comment on this is not clear enough, as it should refer to the staggered unitary rotation Q performed on page 3, which maps the model to a standard XXZ chain.

Thanks for this comment. This was also mentioned by the first referee. We have removed this part totally from this draft and consider it for a related project later.

5) Sec IIIA: As also commented in Report 1, the author should be interested in the density in the original spin variables, whereas the text refers to the transformed fermion density. The author should comment that this is unchanged under the transformation Q, so in practice we are looking at the correct observable.

Thanks the referee for this comment. We add some explanation in this regard. The relation between and boson is also mentioned. Please see lines: 248-250 and 264-268

6) Sec IIIA: the author reports symmetric propagation. I believe this result is correct, however: Fig 2b is for some reason asymmetric, which makes it unnecessarily hard for the reader to grasp the simple result

Thanks the referee for this comment. We correct the figure.

the presentation obscures the main understanding of this result: when viewed in the original fermionic variables (before the transformation Q), the magnon propagates with a dispersion relation . Thus, propagation is not parity symmetric. The only reason why it looks symmetric, is because the initial state is perfectly localized in space, and thus it is a uniform superposition of all momenta. Since the dispersion relation is just shifted in momentum space, it will continue to look symmetric. However, had the author considered a different meaningful initial state, e.g. a magnon wavepacket spread over a width of x>1 lattice sites, this would be nonuniform in momentum space, preferentially selecting momenta around zero. Thus, this wavepacket would move asymmetrically due to DM interactions. I suggest the author to add this explanation and include a simulation showing this effect.

Thanks for this comment. We added a part to consider your suggestion. Please see ines: 357-375 , and Fig.5

7) Sec IIIB: The presentation of linear spin wave theory is misleading. In fact, the LSW treatment is exact in the case presented by the author, because the initial state is composed of a single magnon. What’s more, there is no difference between Jordan-Wigner fermions and hardcore bosons (magnons) in this sector.

Thanks for this comment. We added some more explanation to reconcile the issue. Please see line: 264-267

8) Sec IIIB: The result that long-range interactions produce asymmetric transport even when short-range interactions don’t, seems to be correct. However, the author should improve the analytical understanding of this fact, building on the derived dispersion relation. For example, it is possible to determine the difference between left and right maximum velocities analytically as a function of the parameters, including alpha. I believe that the asymmetry should be present for all alpha’s, but is only appreciable in the plots for small alpha. Fig 3 middle panel suggests this, and the right panel attempts to quantify this, but - I believe - in an incorrect way. In conclusion, if I am right, the description of the main result of the paper should be modified accordingly.

We are really appreciate for this comment. We agree with you, the model renders asymmetry for all $\alpha$ as soon as the DM term turning on. Regarding this we changed the context of paper correspondingly. We have alse tried to wrapping a closed formula, see line:317-325.

9) Sec IIIB: similarly to a previous comment, cf also Report 1, the author should comment that the magnon density is the same as the spin density one is interested in in the original formulation of the problem.

Thanks the referee for this comment. We mentioned it.

Also, above Fig 4 there is a typo: it should be rather than .

Thanks the referee for this comment. We correct it.

10) Sec IIIB: the explanation at the end seems correct and insightful. One can also phrase it in terms of the staggered transformation Q, which does not amount to a simple shift of the dispersion relation when interactions are longer ranged because the lattice is no longer bipartite.

Thanks the referee for this comment. We added this to the text, see lines: 351-355

11) Sec IIIC: I believe this is the weakest point of the paper. As results from the last paragraph of the section, unfortunately the author doesn’t seem to have realized that as long as one considers an initial state in the single particle sector, the interaction is completely immaterial (there is no other particle with which to interact). Indeed, the three plots in Fig 5 look exactly the same despite the widely different values of the interaction. To study the effect of interactions, the author should consider initial state with multiple particles, such as those considered in Ref. 25. I suggest the author to do this, or to remove Fig 5 and the discussion of interactions altogether from the paper to avoid any misinterpretations by the readers. If this is not correct, the author should anyway discuss this point to avoid confusion.

We are very thankful to bring us this misunderstanding. We choose another initial state to consider truly the effect of many-body. Please see section C and Fig.6.

12) Sec IIID: apart from Refs. 7,8,9,11,12, the author should cite Phys. Rev. Research 2, 012041(R) for the effect of long-range interactions on the entanglement dynamics. Note that following comment 11, the dynamics of entanglement would become truly more interesting in the manybody sector, as entanglement entropy would be created not only by the uncertainty in the position of the particle across the cut (and hence bounded by log2 as in Fig 8) but also by genuine quantum correlations shared by the particles.

Thanks for this comment. We added the mentioned reference. Regarding EE, as the first referee also pointed it out, we added a few more lines to connect it with possible confinement of elementary excitations. Please see line: 424-428 and 443-448.

---

## Round 2 · Referee Report · Anonymous (Referee 2) · 2022-2-14

Report

The author addressed my previous comments and improved the manuscript, clarifying the previous confusion on the interpretation of the results.
I believe the manuscript is now suitable for publication in SciPost Physics Core (although the text would still benefit from improving the language).
As a last comment, I recommend reporting in sec. IIID the initial state considered (I interpreted it is the same as in sec. IIIC, but this is written neither in the text nor in the captions of Fig 7, 8).

---

## Round 2 · Referee Report · Anonymous (Referee 1) · 2022-3-17

Strengths

  1. The paper addresses an important and timely topic.
  2. The paper is well written and interesting.
  3. The microscopic models and theoretical methods used to study them are well explained.

Weaknesses

No significant weakness.

Report

The author has taken into account the previous criticisms in an appropriate manner. I suggest publication in Scipost Physics Core.

---

## Round 2 · Author Response

We thank the anonymous referee for reading the paper carefully and providing thoughtful comments, many of which have resulted in changes to the revised version of the manuscript. Below you can find, reply to the comments.

---

## Round 2 · List of Changes

Warnings issued while processing user-supplied markup:

  • Inconsistency: plain/Markdown and reStructuredText syntaxes are mixed. Markdown will be used.
    Add "#coerce:reST" or "#coerce:plain" as the first line of your text to force reStructuredText or no markup.
    You may also contact the helpdesk if the formatting is incorrect and you are unable to edit your text.

Report 1: I have reviewed the paper entitled “Asymmetric transport in long-range interacting chiral spin chains”, by Javad Vahedi. The paper investigates the effect of Dzyaloshinskii-Moriya (DM) interactions in one-dimensional spin chains, with power-law decaying interactions, onto the spreading of correlations after a local quench. In principle, the problem is interesting and relevant to the readership of Scipost, but I am not sure that all presented results are valid, and there are major inconsistencies in the paper. Therefore, I do not think the paper should be published, at least not without major revisions.

1) First of all, the paper is poorly written and contains many typos and grammatical errors. Before any resubmission, I strongly suggest the author to ask a fluent english-speaking collegue to read the paper carefully.

Thanks the referee for this comment. We try to polish and resolve typos in the text.

2) When mapping the spin chain onto free fermions (III.A), the quantities of interest are still the spin correlations. Instead, the fermion correlations are presented in Fig. 2. The author should instead consider the (initial, i.e. unrotated) spin variables, and study the evolution of the mean magnetization, and possibly also the spin-spin correlations.

Thanks the referee for this comment. We correct it. Please see page 3, right column, end of the second paragraph (—> Note that connection with …) and see page 5, left column, third paragraph line 5 (—> Note that connection with original …)

3) In III.B, the use of LSW approximation should be further discussed. Around which mean-field state is it performed? Is the criterion valid at all sites and at which times? Furthermore, here also, one is primarily interested in the spin variables, and not in the bosonic variables (see Fig. 4). The author should discuss the relation between them.

Thanks the referee for this comment. We add some explanation in this regard. Please see subsection B on page 4, end of the first paragraph (—-> Moreover, for the initial …) and end of the second paragraph ( —-> Considering the initial ordering …)

4) In the caption of Fig. 5, the Ising model cannot be simply . A rescaling has been made, and should be explicited by the author.

Thanks the referee for this comment. We change section 4 in the revised version and correspondingly Fig.5 is also changed.

5) The different regimes of interaction range are not convincing. For instance, one clearly sees the asymmetry of the dispersion relation for all when . Why would it have no consequence on the spreading of correlations? Furthermore, the asymmetry of the mutual information (III.D, Fig. 7), is clearly present for all . This is in contradiction with the main claims of the paper.

Many thanks for the comment. That is totally true, asymmetry exits for all $\alpha$ as soon as $D\neq 0$. This whys you can see the asymmetry in III.D, Fig. 7. But for a fixed $D$, say $D=1.0$ we have considered in the paper, a really long-range $\alpha$ needs to see an appreciable effect on spreading.

6) The author finds that the half-chain entropy is reduced at long times in the presence of DM interactions. Why is it so? Does the author have a physical explanation?

Many thanks for the comment. We add some explanation in this regard. Please see subsection D on page 6, end of the second paragraph (—> This may be explained ….) and end of the third paragraph ( —> Another explanation ….)

7) In the appendix, the author investigates the level statistics of the models, but this is not discussed in the main text. What is the added value of this study?

Thanks for this comment. We have removed this part totally from this draft. We will consider it for a related project later.

In conclusion, although the topic is a priori interesting, I have many doubts on both the technical validity of the results, and on their consistent interpretation. Therefore, I cannot recommend the publication without major revisions, and an almost complete rewriting of the paper.

=============

=============

=============

Report 2:

My overall assessment of the paper is positive, as the author has considered a truly interesting, theoretically well motivated and experimentally relevant problem, which to the best of my knowledge has not been addressed before in this form. The presentation is sound and previous literature is mostly reported and correctly cited. Above all, to further credit of the author, I believe the presented main result - i.e. the discovery of asymmetric particle transport due to the combination of long-range and DM interactions - is correct, although the interpretation of some numerical and analytical results seems incorrect or rather incomplete (see below), such that the whole presentation sounds a bit misleading. For these reasons, I recommend publication of this work in SciPost Physics only after the author carefully amends the various issues I discuss below. Note that some of them address points also raised by a previous Referee in Report 1.

As in Report 1, I also recommend revision of the grammar and misprints in the text. Thanks the referee for this comment. We have tried to polish the text and resolve typos in the text.

2) The author writes two times that the light-cone ceases to exist only for alpha<1 (intro and page 4). This is misleading, as the light-cone effect does not exist as soon as the maximum propagation velocity becomes infinite, i.e., for alpha<2 (as the author recognizes). I recommend to modify the phrasing.

Thanks the referee for this comment. We correct it.

3) page 2 second column: Ref 7 is cited as an experiment, while it is a numerical simulation.

Thanks the referee for this comment. We correct it.

4) the second appendix reports the study of level statistics. While its utility is questionable, as the author always works in the single-particle sector (see below), the reported results suggest integrability of the nearest-neighbor interacting XXZ+DM spin chain. The author’s comment on this is not clear enough, as it should refer to the staggered unitary rotation Q performed on page 3, which maps the model to a standard XXZ chain.

Thanks for this comment. This was also mentioned by the first referee. We have removed this part totally from the draft and consider it for a related project later.

5) Sec IIIA: As also commented in Report 1, the author should be interested in the density in the original spin variables, whereas the text refers to the transformed fermion density. The author should comment that this is unchanged under the transformation Q, so in practice we are looking at the correct observable.

Thanks the referee for this comment. We add some explanation in this regard. Please see page 3, right column, end of the second paragraph (—> Note that connection with …).

6) Sec IIIA: the author reports symmetric propagation. I believe this result is correct, however: Fig 2b is for some reason asymmetric, which makes it unnecessarily hard for the reader to grasp the simple result Thanks the referee for this comment. We correct the figure.

the presentation obscures the main understanding of this result: when viewed in the original fermionic variables (before the transformation Q), the magnon propagates with a dispersion relation . Thus, propagation is not parity symmetric. The only reason why it looks symmetric, is because the initial state is perfectly localized in space, and thus it is a uniform superposition of all momenta. Since the dispersion relation is just shifted in momentum space, it will continue to look symmetric. However, had the author considered a different meaningful initial state, e.g. a magnon wavepacket spread over a width of x>1 lattice sites, this would be nonuniform in momentum space, preferentially selecting momenta around zero. Thus, this wavepacket would move asymmetrically due to DM interactions. I suggest the author to add this explanation and include a simulation showing this effect.

Thanks for this comment. We added a part to consider your suggestion. Please see page 5, paragraph 4 (—> Before closing this part …) and Fig.5

7) Sec IIIB: The presentation of linear spin wave theory is misleading. In fact, the LSW treatment is exact in the case presented by the author, because the initial state is composed of a single magnon. What’s more, there is no difference between Jordan-Wigner fermions and hardcore bosons (magnons) in this sector.

Thanks the referee for this comment. We add some explanation in this regard. Please see subsection B on page 4, end of the first paragraph (—-> Moreover, for the initial …) and end of the second paragraph ( —-> Considering the initial ordering …)

8) Sec IIIB: The result that long-range interactions produce asymmetric transport even when short-range interactions don’t, seems to be correct. However, the author should improve the analytical understanding of this fact, building on the derived dispersion relation. For example, it is possible to determine the difference between left and right maximum velocities analytically as a function of the parameters, including alpha. I believe that the asymmetry should be present for all alpha’s, but is only appreciable in the plots for small alpha. Fig 3 middle panel suggests this, and the right panel attempts to quantify this, but - I believe - in an incorrect way. In conclusion, if I am right, the description of the main result of the paper should be modified accordingly.

We are really appreciate for this comment. We agree with you, the model renders asymmetry for all $\alpha$ as soon as the DM term turning on. Regarding this we changed the context of paper correspondingly. We have also tried to find a formula, see page 5, left column, first paragraph (—> Analytically, on can …).

9) Sec IIIB: similarly to a previous comment, cf also Report 1, the author should comment that the magnon density is the same as the spin density one is interested in in the original formulation of the problem. Thanks the referee for this comment. We mentioned it. Please see page 5, left column, third paragraph line 5 (—> Note that connection with original …)

Also, above Fig 4 there is a typo: it should be rather than .

Thanks the referee for this comment. We correct it.

10) Sec IIIB: the explanation at the end seems correct and insightful. One can also phrase it in terms of the staggered transformation Q, which does not amount to a simple shift of the dispersion relation when interactions are longer ranged because the lattice is no longer bipartite.

Thanks the referee for this comment. We added this to the text, please see page 5, left column, end of the third paragraph (—> This can be …)

11) Sec IIIC: I believe this is the weakest point of the paper. As results from the last paragraph of the section, unfortunately the author doesn’t seem to have realized that as long as one considers an initial state in the single particle sector, the interaction is completely immaterial (there is no other particle with which to interact). Indeed, the three plots in Fig 5 look exactly the same despite the widely different values of the interaction. To study the effect of interactions, the author should consider initial state with multiple particles, such as those considered in Ref. 25. I suggest the author to do this, or to remove Fig 5 and the discussion of interactions altogether from the paper to avoid any misinterpretations by the readers. If this is not correct, the author should anyway discuss this point to avoid confusion.

We are very thankful to bring us this misunderstanding. We choose another initial state to consider truly the effect of many-body. Please see section C and Fig.6.

12) Sec IIID: apart from Refs. 7,8,9,11,12, the author should cite Phys. Rev. Research 2, 012041(R) for the effect of long-range interactions on the entanglement dynamics. Note that following comment 11, the dynamics of entanglement would become truly more interesting in the manybody sector, as entanglement entropy would be created not only by the uncertainty in the position of the particle across the cut (and hence bounded by log2 as in Fig 8) but also by genuine quantum correlations shared by the particles.

Thanks for this comment. We added the mentioned reference. Regarding EE, as the first referee also pointed it out, we added a few more lines to connect it with possible confinement of elementary excitations. Please see subsection D on page 6, end of the second paragraph (—> This may be explained ….) and end of the third paragraph ( —> Another explanation ….)

---

## Editorial Decision

published